# Twins’ Macular Pigment Optical Density Assessment and Relation with SCARB1 Gene Polymorphism

**DOI:** 10.3390/genes14010125

**Published:** 2023-01-02

**Authors:** Edita Kunceviciene, Ruta Mockute, Aiste Petrauskaite, Brigita Budiene, Alina Smalinskiene, Ieva Zvykaite, Rasa Liutkeviciene

**Affiliations:** 1Institute of Biology Systems and Genetics Research, Lithuanian University of Health Sciences, Tilzes 18, 47181 Kaunas, Lithuania; 2Department of Ophthalmology, Lithuanian University of Health Sciences, Eiveniu 2, 50161 Kaunas, Lithuania; 3Neuroscience Institute, Lithuanian University of Health Sciences, Eiveniu 2, 50161 Kaunas, Lithuania

**Keywords:** monozygotic twins, dizygotic twins, macular pigment optical density, SCARB1

## Abstract

The aim of the study: to assess the influence of genetic and environmental factors using twin studies and evaluate the associations of SCARB1 gene variants (rs11057841) with AMD and MPOD. Material and methods: a total of 108 healthy twins (56 MZ and 52 DZ twins) were tested in this study. The MPOD was measured using the one-wavelength reflectometry method. Fundus reflectance (Visucam 500, reflectance of a single 460 nm wavelength) was used to measure the MPOD levels, MPOD parameters including max and mean optical density (OD), and area and volume. Real-time polymerase chain reaction was used to detect single nucleotide polymorphisms. Results: we detected a positive correlation of MPOD in the right and left eyes in MZ twin pairs (r = 0.830 and r = 0.860, respectively) (*p* < 0.0001) and a negative correlation of MPOD in the right and left eyes in DZ twin pairs (r = 0.314 and r = 0.408, respectively) (*p* < 0.05). The study was able to identify statistically significant differences in mean MPOD values in the right and left eyes between subjects with a wild-type CC genotype and a CT genotype with a risk allele. A decrease in the mean MPOD value was observed in group II with a CT genotype (0.110 d.u.) compared with the CC genotype (0.117 d.u.) in the right eye (*p* = 0.037) and in the left eye with a CT genotype (0.109 d.u.) compared with a CC genotype in the subjects (0.114 d.u.) (*p* = 0.038). In the right eye, in group II (0.101–0.128 d.u.), those with a CT genotype (*n* = 6) with one risk allele had a statistically significantly lower (0.110 d.u.) mean average MPOD value compared with those with a wild-type CC genotype (*n* = 25) (0.117 d.u.) (*p* = 0.037). Conclusion: this twin study showed a strong heritability of the retina pigment, which was 86% prevalent in Lithuania. Individuals with a CT genotype of the SCARB1 rs11057841 with a risk allele had statistically significantly lower mean MPOD values in both eyes compared to subjects with a wild-type CC genotype.

## 1. Introduction

Macular pigment (MP) absorbs the light of short wavelengths acting as a selective pre-receptor filter [1]. MP can protect the retina from blue light by absorbing high-energy short-wavelength light (450–500 nm); therefore, a low-density macular pigment may represent a risk factor for age-related macular degeneration (AMD) [2]. AMD is a disease that affects the area of retinal macula and causes the gradual loss of central vision [3]. The attenuation of blue light by macular pigment is a measure of the macular pigment optical density (MPOD) and is related to the amount of concentration, pathlength, and area of lutein and zeaxanthin in the macula [4].

Scientists have found different macular pigmentations in Europeans and Asians; for example, Caucasians and African Americans have a lower macular pigment optical density compared with South Asian Indians and Hispanics [5,6,7,8]. It is difficult or even impossible to compare MPOD levels among different studies because of the employment of different methodologies or study protocols.

AMD is a disease of multifactorial etiology, affected by genetic, environmental, and lifestyle factors, so it is very important to find out what factors affect the development and progression of AMD [3]. To what extent genetic factors determine the formation of macular density, not much research has been conducted, and macular density can also vary in other populations. With the excellent opportunity to apply the twin method, we performed an assessment of genetic and environmental factors for macular pigment.

Twin studies allow us to estimate the overall influence of genes for disease or any qualitative or quantitative trait. The twin method assumes that DZ (dizygotic) and MZ (monozygotic) twins alike are influenced by environmental differences, but while DZ twins share only about half of their genes, MZ twins share the same genes [9].

Some AMD-related genes and environmental factors (e.g., smoking) interact and have a significant impact on the onset and development of AMD. The role of the lipid pathway has been implicated in the pathogenesis mechanism of AMD due to the accumulation of lipids in the Bruch membrane and drusen. Drusen are a distinguishing feature of AMD, so efforts have been made to find out which genes are involved in lipid metabolism and regulate these processes. The SCARB1 gene on the long shoulder of chromosome 12 (12q24.31) encodes the 509 amino acid SRBI protein, which mediates the transport of cholesterol between cells and high-density (HDL) cholesterol and is also involved in the metabolism of vitamin E and lutein. Lutein is a component of macular pigment and is essential for the visual process, and vitamin E has protective properties against cellular oxidative damage, so genetic alterations in the SCARB1 gene may be relevant in the study of AMD’s pathogenesis [3]. Changes in macular pigment are determined by measuring optical density [10]. The literature suggests that a decrease in macular pigmentation results in poorer retinal protection and may lead to the onset of AMD [11].

There is still a lack of scientific literature on polymorphisms and genetic alterations in the SCARB1 gene and on its interactions with AMD and MPOD. Analyses of SCARB1 gene polymorphisms may help identify new gene variants that are significant in the pathogenesis of AMD.

The purpose of our study was to assess the MPOD in pairs of MZ and DZ twins and evaluate the associations of SCARB1 gene variants (rs11057841) with AMD and MPOD.

## 2. Methods

### 2.1. Ethics Statement

Permission (BE-2-41/2020) to undertake the study was obtained from the Kaunas Regional Biomedical Research Ethics Committee. The investigation was conducted in the Department of Ophthalmology at the Lithuanian University of Health Sciences.

### 2.2. Study Samples

#### 2.2.1. MPOD Studies Using the Twin Method

MPOD characteristics were studied in 108 twins registered at the Twin Center of the Lithuanian University of Health Sciences (LUHS). Exclusion criteria for the study group: (1) age ≤ 18; (2) refusal to participate in the investigation; and (3) lens opacity (nuclear, cortical, or posterior subcapsular cataract), keratitis, acute or chronic uveitis, glaucoma, or diseases of the optic nerve.

#### 2.2.2. Study of Single Nucleotide Polymorphism in the SCARB1 Gene

Two hundred and seventy-two individuals participated in the SCARB1 rs11057841 polymorphism study. The study sample consisted of a control group—171 healthy subjects and an AMD group consisting of 101 subjects; 61 individuals had early AMD and 40 individuals had late AMD. The control sample of the study consisted of two subgroups: (1) ophthalmologically healthy twins registered at the LUHS Twin Center (in the years 2004–2014 and 2016–2017). In the gene polymorphism studies, these twins were selected: (1) the first-born person from a pair of twins and (2) a group of healthy individuals from the LUHS, the Department of Ophthalmology. The inclusion criteria of the samples mentioned before were also followed.

### 2.3. MPOD Measurement

This study investigated visual acuity, transparency of the cornea and lens, and the fundus in pairs of monozygotic (MZ) and dizygotic (DZ) twins. The corneal and lenticular transparencies were assessed with bio-microscopy. Uncorrected and best-corrected visual acuities were evaluated using Landolt rings (C optotypes) with Snellen’s test types at a 5 m distance from the chart. A slit-lamp, with the illumination source positioned at a 45° angle and the light beam set to a 2 mm width, was used to examine the lenses. Then, funduscopy was performed with an ophthalmoscope of the direct monocular type and a slit-lamp using a double aspheric lens of +78 diopters. For a detailed macula analysis, stereoscopic color fundus photographs of the macula centered at 45° from the fovea were obtained with a Visucam NM digital camera (Carl Zeiss Meditec AG, Jena, Germany).

The MPOD was objectively evaluated using a Visucam 500 digital fundus camera, which measured the reflectance of the blue light close to the macular pigment’s area of maximum absorption.

The MPOD was measured at an angle of 30° from the fovea where the highest macular pigment density was detected. The MPOD and its spatial distribution using a Viscam 500 were an optional macular pigment density module using the reflectance of a single 460 nm wavelength based on a single blue-reflection fundus image. To approximate the reflectance of the fundus in the absence of MP, shading correction was used. This was based on a three-dimensional parabolic function automatically fitted to the fundus reflectance at the peripheral locations. The participant was positioned in front of the fundus camera and instructed to look at a target inside. All images were collected by the same technician, under the same light conditions, with the same flash intensity, and after mydriasis. The subjects had to look at a target inside the fundus camera. The fundus was illuminated with a monochromatic blue light. Four MPOD parameters were measured during the study: the maximum optical density (MPOD measured at the peak); mean OD (mean MPOD within the measurement area); area (area where macular pigment could be detected); and volume (sum of all optical densities, as recommended by the manufacturer) [12].

### 2.4. Twin Method

The heritability of MPOD was assessed using the twin method. The narrow-sense heritability coefficient (h^2^) was calculated in the interclass MZ and dizygotic (DZ) groups, based on Pearson correlations (r), according to the formula: h^2^ = 2 × (r_MZ_ − r_DZ_). Heritability indicates the extent to which additive genes contribute to MPOD.

### 2.5. DNA Extraction

Peripheral blood samples were collected from each individual in ethylenediaminetetraacetic (EDTA) tubes for DNA extraction. DNA was extracted from leukocytes using a reagent kit (NucleoSpin Blood L Kit, Macherey & Nagel, Düren, Germany). DNA samples from one member of each MZ pair were used for genotyping [13].

### 2.6. Verification of Zygosity

Zygosity was determined using a DNA test. A polymerase chain reaction set (AmpFlSTR^®^ Identifiler^®^, Applied Biosystems, Foster City, CA, USA) was used to amplify short tandem repeats; 15 specific DNA markers were used for the comparison of genetic profiles: D8S1179, D21S11, D7S820, CSF180, D3S1358, TH01, D13S317, D16S539, D2S1338, D19S433, vWA, TROX, D18S51, D5S818, and amelogenin.

### 2.7. Genotyping

The genotyping of two SNPs of the SCARB1 (rs11057841) gene were performed for 272 individuals.

RT-PCR (Applied Biosystems, Foster City, CA, USA) was used for detecting the SNPs by using allelic discrimination. The cycling program started with heating for 10 min at 95 °C, followed by 40 cycles of 15 s at 95 °C and 1 min at 60 °C [13].

### 2.8. Statistical Analysis

A statistical data analysis was performed using the IBM SPSS Statistics computer program. Data are presented as actual numbers (percentages), maximum and minimum values, and standard deviation. SCARB1 genotyping was performed with the consideration of appropriate sample size estimation and desirable precision. The distribution of SCARB1 gene variants in our AMD and control groups was evaluated according to the Hardy–Weinberg law. The χ2 criterion was used to compare the homogeneity of the distribution of polymorphism genotypes between patients with AMD and the control group. Student’s T criterion was used to compare MPOD characteristics between different groups of values with the genotype with and without the risk allele. The study calculated the intraclass correlation of MPOD values between monozygotic and dizygotic twin pairs.

## 3. Results

### 3.1. Influence of Genetic and Environmental Factors on Macular Pigment Density Using the Twin Study Method

There were 108 individuals who participated in the study, of which 56 were MZ twins and 52 were DZ twins. Among the MZ twins, there were 40 females (71.4%) and 16 males (28.6%). Among the DZ twins, there were 38 females (76%) and 14 males (26.9%). The characteristics of the genders and ages of the MZ and DZ twins are given in Table 1.

The study evaluated the influence of genetic and environmental factors on macular pigment optical density by comparing pairs of monozygotic (MZ) and dizygotic (DZ) twins. The study measured four MPOD characteristics (volume, area, and maximum and mean MPOD values) in MZ and DZ twins.

The intraclass correlation of MPOD values between monozygotic and dizygotic twin pairs was calculated during the study (Table 2). A higher correlation between the mean of the MPOD values in both eyes was found in MZ twins. The correlation coefficients (r) of the MPOD for right and left eyes in MZ twin pairs were r = 0.830 and r = 0.860, respectively (*p* < 0.0001). The correlation coefficients for DZ twin pairs in the right and left eyes were not significant; r = 0.314 and r = 0.408, respectively (*p* > 0.05).

The MPODs for twin 1 versus twin 2 for the mean MPOD values are shown in Figure 1. The intrapair correlations of the MPOD in both eyes in the twin pairs were significantly higher in MZ twin pairs compared with DZ twin pairs, proving that genetic factors are more responsible for macular pigment density (83–86%) compared to environmental factors (14–17%).

### 3.2. Distribution of SCARB1 rs11057841 Genotypes in AMD and Control Groups

A total of 272 persons participated in the study, during which two groups of subjects were formed. The control group consisted of healthy individuals, and the study group consisted of patients with early or late forms of AMD. The characteristics of the subjects are presented in Table 3. There were 171 people in the control group, of which 50 (29.2%) were males and 121 (70.8%) were females. The average age of this group was 56.86. The group of subjects with AMD consisted of 101 patients, of which 61 people had early AMD, and 40 people had late AMD. The research group consisted of 77 females (76.2%) and 24 (23.8%) males; the average age was 60.85. A statistically significant difference in the mean age of females was observed between the AMD group (60.90) and the control group (56.52) (*p* = 0.004).

The frequency of SCARB1 gene rs11057841 polymorphism was evaluated in patients with permanent macular degeneration and in the control group (Table 4). Both in the group of patients with AMD and in the group of healthy individuals, the distribution of genotypes corresponded to the Hardy–Weinberg law (*p* > 0.05). A similar distribution of genotypes was observed in both groups, but the TT genotype was slightly more frequent in the AMD group (TT = 3%). No statistically significant distribution of genotypes (*p* = 0.508) and alleles (*p* = 0.847) was found between healthy and diseased individuals.

Table 5 shows the mean, maximum and minimum values, and standard deviation of the MPOD characteristics of both eyes. The mean differences between the right and left eyes were observed—the volume, maximum, and mean MPOD values were greater in the right eye, and the area was greater in the left eye.

In the right eye, in group II (0.101–0.128 d.u.), those with a CT genotype (*n* = 6) with one risk allele had a statistically significantly lower (0.110 d.u.) mean MPOD value compared with those with a wild-type CC genotype (*n* = 25) (0.117 d.u.) (*p* = 0.037) (Table 6). In the left eye, there was also a significant decrease in the mean MPOD value (0.109 d.u.) between those with a single allele of the CT genotype (*n* = 6) and those with a wild-type CC (*n* = 26) genotype (0.114 d.u.) in group II (0.101- 0.125 d.u.) (*p* = 0.038) (Table 6).

## 4. Discussion

We found that there have been only a few studies that examined MPOD data in twins [14,15], but to our knowledge, none of these studies used a Visucam 500 to measure macular pigment optical density. We estimated the heritability of MPOD in MZ and DZ twin pairs. Our study showed that the heritability of MPOD was 0.86.

Another study investigated 150 twin pairs, aged 18–50 years old. To psychophysically measure the MP optical density, the heterochromatic flicker photometry (HFP) and fundus autofluorescence (AF) imaging methods were used. This twin study demonstrated that MP optical density is generally determined by genetic background, which is reflected in the HFP and AF heritability estimates of 0.67 and 0.85, respectively [15].

Humans cannot synthesize MP and must obtain carotenoids through a diet of fruit, vegetables, and egg yolk [12]. MP as an antioxidant protects the retina from free radicals formed by oxidative stress. Consequently, MP might play a protective role against degenerative eye diseases such as age-related macular degeneration. In order to study the influence of genes on the formation of MP, it is important to study the amount of MP in twins. The high repeatability and reliability of the assessment are especially important in studies investigating how MPOD is influenced by the diet and/or nutritional supplements of lutein and meso-Zeaxanthin or disease processes [16,17].

Up to now, no analogous studies have been performed to analyze SCARB1 gene polymorphism in various forms of AMD, and therefore, little data are available on the involvement of this gene and its polymorphisms in ocular disease processes, aging, and the pathogenesis of AMD. However, an analysis of the scientific literature suggests possible associations between the SCARB1 gene and the pathogenesis of AMD, as the gene encodes HDL receptors, which play an important role in cholesterol and lutein metabolism and may be involved in cholesterol-containing drusen.

In the study, we tried to find out whether the polymorphism of the SCARB1 gene rs11057841 is related to the optical density characteristics of macular pigment. A separate group of subjects with four MPOD characteristics in both eyes was examined. All four characteristics (volume, area, and maximum and mean MPOD values) in both eyes were compared between individuals with the wild-type CC genotype and between carriers of the T allele between the three different groups, with increasing values. Statistically significant (*p* < 0.05) differences in mean MPOD values in the right and left eyes were found. The average mean MPOD values were found to be lower among subjects with the CT genotype compared with subjects with the CC genotype.

An association between MP and lipoproteins has been established because the major components of MP (lutein and zeaxanthin) are transported via lipoproteins. The SCARB1 gene is involved in lipid transport, and mutations and alterations in this gene may result in impaired lutein and zeaxanthin transport and lower MPOD values [17]. Based on studies by other researchers, SCARB1 rs11057841 polymorphism has been shown to be possibly associated with increased serum lutein levels, which may be due to the imbalance of rs11057841 with rs10846744. A positive family history has also been found to be positively correlated with serum lutein levels and rs11057841 [18,19].

A study conducted in the French population showed that the distribution of SCARB1 rs5888 polymorphism genotypes differed significantly (*p* < 0.01) between patients with AGDD and control subjects; those with the CT genotype compared to the CC genotype had a higher risk of AGDD. Epidemiological studies in Caucasian populations have shown that rs5888 is associated with increased HDL cholesterol and lower LDL cholesterol and is also associated with an increased risk of coronary heart disease in males [19].

Our study was able to determine the importance of genetic factors for macular pigment and calculated several statistically significant values that indicated a possible association of the SCARB1 gene rs11057841 with MPOD characteristics. Although no significant differences could be obtained to demonstrate the association of the SCARB1 rs11057841 polymorphism with AMD, an analysis of the scientific literature suggests that the SCARB1 gene rs11057841 polymorphism may be more closely related to pathogenesis and between different age groups and health conditions. It is advisable to use several polymorphisms of the same gene for research or to simultaneously study several similar genes and to monitor their interaction with each other.

## 5. Conclusions

This twin study showed the strong heritability of the retina pigment, which was 86% prevalent in Lithuania. Individuals with a CT genotype of the SCARB1 rs11057841 gene with a risk allele had statistically significantly lower mean MPOD values in both eyes compared with subjects with a wild-type CC genotype.

## Figures and Tables

**Figure 1 genes-14-00125-f001:**
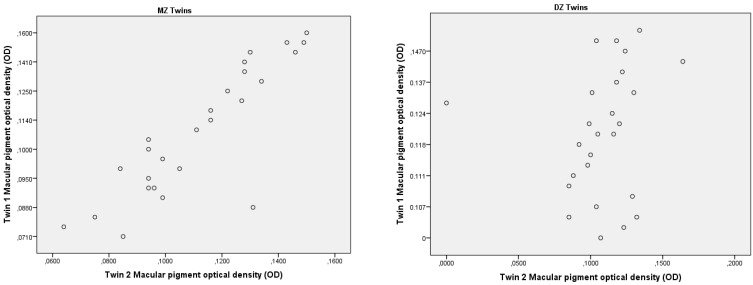
Scatter plots of MPOD in each twin’s right eye, twin 1 versus twin 2, in 56 MZ and 52 DZ twin pairs. MZ—monozygotic twins; DZ -dizygotic twins; OD—right eye.

**Table 1 genes-14-00125-t001:** Demographic characteristics of the study population.

Characteristic	Results	
	MZ Twins	DZ Twins
Males, *n* (percent)	16 (28.6)	14 (26.9)
Females, *n* (percent)	40 (71.4)	38 (76)
Age, min–max (median)	25–66 (40)	18–63 (39)

*n*—samples; MZ—monozygotic twins; DZ—dizygotic twins; min–max—minimum and maximum values.

**Table 2 genes-14-00125-t002:** Correlation of MPOD values between MZ and DZ twins.

	Correlation Coefficient (r)	Effect of Genetic Factors (Percent)	Effect of Environmental Factors (Percent)	*p*
**MZ twins (*n* = 56)**				
**OD**	0.830	83	17	<0.0001
**OS**	0.860	86	14	
**DZ twins (*n* = 52)**				
**OD**	0.314	31.4	68.6	>0.05
**OS**	0.408	40.8	59.2	

OD—right eye; OS—left eye; MZ—monozygotic; DZ—dizygotic; *p*—the significance level calculated. The differences are considered statistically significant when *p* < 0.05.

**Table 3 genes-14-00125-t003:** Comparison of mean optical density and genotypes of SNP rs11057841 in SCARB1 gene.

Samples	AMD Group *n* = 101	Controls *n* = 171	*p*
**Gender**			
**Males, *n* (percent)**	24 (23.8)	50 (29.2)	0.327
**Females, *n* (percent)**	77 (76.2)	121 (70.8)	
**Average age (min;max)**			
**Males (years)**	60.71 (45;68)	57.68 (46;70)	0.200
**Females (years)**	60.90 (42;92)	56.52 (45;75)	0.004

*p*—level of significance; differences are considered statistically significant when *p* < 0.05.

**Table 4 genes-14-00125-t004:** Comparison of mean optical density and genotypes of SNP rs11057841 in SCARB1 gene.

Genotype	Frequency of Alleles and Genotypes of SCARB1 SNP rs11057841 (Percent)						
	AMD Group *n* (percent), *n* = 101	χ^2^	*p* Value HWE*	Controls *n* (percent), *n* = 171	χ^2^	*p* Value HWE*	*p*
**CC**	80 (79.2)	2.239	0.135	134 (78.4)	0.029	0.866	0.508
**CT**	18 (17.8)			35 (20.5)			
**TT**	3 (3.00)			2 (1.2)			
**Allele**							
**C**	178 (88.1)	ꟷ		303 (88.6)	ꟷ		0.866
**T**	24 (11.9)			39 (11.4)			

AMD—age-related macular degeneration; *p* value HWE*—significance level according to the Hardy–Weinberg law; *p*—level of significance. Differences are considered statistically significant when *p* < 0.05.

**Table 5 genes-14-00125-t005:** Left and right eye MPOD characteristics data.

OD MPOD Characteristics				
MPOD Characteristics	Volume (Density Units)	Area (Density Units)	Max MPOD Value (Density Units)	Mean MPOD Value (Density Units)
**Min value**	3546	6729	0.203	0.064
**Max value**	12717	97874	0.457	0.152
**Average**	7732.26	66,603. 34	0.341	0.115
**SD**	1959.165	14,424.762	0.052	0.018
**OS MPOD characteristics**				
**Min value**	1788	6488	0.173	0.050
**Max value**	11,619	92,909	0.430	0.157
**Average**	7490.170	66,911.750	0.333	0.111
**SD**	1884.328	13,929.306	0.056	0.020

OD—right eye; OS—left eye; SD—standard deviation.

**Table 6 genes-14-00125-t006:** Comparison of mean optical density and SCARB1 SNP rs11057841 genotypes in the AMD group.

Mean MPOD (Density Units)				
Right Eye				
Group I (<0.101)				
Genotype	n	SD	Mean	*p*
**CC**	13	0.011	0.088	0.279
**CT**	1	–	0.075	
**Group II (0.101–0.128)**				
**CC**	25	0.007	0.117	0.037
**CT**	6	0.005	0.110	
**Group III (>0.128)**				
**CC**	13	0.008	0.140	0.730
**CT**	2	0.009	0.138	
**Left eye**				
**Group I (<0.101)**				
**CC**	14	0.015	0.083	0.579
**CT**	2	0.007	0.089	
**Group II (0.101–0.125)**				
**CC**	26	0.005	0.114	0.038
**CT**	6	0.004	0.109	
**Group III (>0.125)**				
**CC**	15	0.009	0.134	0.742
**CT**	1	-	0.137	

SDstandard deviation; *p*—level of significance. Differences are considered statistically significant when *p* < 0.05.

## Data Availability

The datasets used and/or analyzed during the current study are available from the corresponding author on reasonable request.

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
