# Peer review of "Twins’ Macular Pigment Optical Density Assessment and Relation with SCARB1 Gene Polymorphism"

_genes, 2023, doi:10.3390/genes14010125_

Round 1
Reviewer 1 Report
the manuscript is well written ,only the references should be improved to add recent references,
the most recent one was on 2018
Author Response
Dear Reviewer,
we greatly appreciate the revision of our manuscript. We would like to take this opportunity to express our sincere gratitude to the reviewers and editors who helped improve the manuscript. We would also like to thank you for the opportunity to resubmit a revised manuscript. We hope that the revised manuscript will be acceptable for publication in your journal. Please also find below our point-by-point submitted file to your comments.

Reviewer 2 Report
MPOD in pairs of MZ and DZ twins and evaluate the associations of SCARB1 gene variants with AMD and MPOD are determined in this study. I believe, to publish the paper in such a reputed journal needs to be revised carefully as mentioned below:
1. Better to emphasize the MPOD method in section 2 and other measurement-related aspects.
2. Shift the other unrelated data (not related to the current measurement) to the introduction section.
3. Major reorganization is required in the manuscript. Like, as merging of result and discussion section for proper exploration.
4. Conclusions over study are not sufficient. Better if include the major conclusions of the study.
5. An author can include a comparative study of this in the discussion section. As various other studies are available in open source.
Author Response

(The authors gave the same response as above.)

Reviewer 3 Report
The study by Kunceviciene et al measured macular pigment optical density (MPOD) in monozygotic and dizygotic twins and evaluated the associations of SCARB1 gene polymorphism with MPOD and AMD.
Overall, the manuscript lacks clarity and does not read well. Will also need editing in terms of sentence structure, grammar and typo errors.
Major concern
The heading mentions about AMD; however, I can hardly find any information pertaining to AMD in the entire manuscript.
Introduction – Not coherent and appears vague. The authors should focus on the topic rather providing information in bits and pieces.
Methods - No information on the study population as well as Inclusion and exclusion criteria? Cannot find any information in relation to AMD (definition, stage, mode of identification??)
Statistical analysis- No information on sample size estimation.
Results – With the exception of table 5, the entire result section is related to macular pigment. I cannot find any information on AMD?
Figures- Are the given figures providing additional information?? The legends for figures mention about …twin pairs in myopia group. What does that refer to?
Discussion- Not focussed on the topic. Too much generic information on macular pigment is included.
Author Response

(The authors gave the same response as above.)

Round 2
Reviewer 2 Report
No more comments.